# TXI (Texture and Color Enhancement Imaging) for Serrated Colorectal Lesions

**DOI:** 10.3390/jcm11010119

**Published:** 2021-12-27

**Authors:** Toshihiro Nishizawa, Osamu Toyoshima, Shuntaro Yoshida, Chie Uekura, Ken Kurokawa, Munkhbayar Munkhjargal, Miho Obata, Tomoharu Yamada, Mitsuhiro Fujishiro, Hirotoshi Ebinuma, Hidekazu Suzuki

**Affiliations:** 1Gastroenterology, Toyoshima Endoscopy Clinic, Tokyo 157-0066, Japan; nisizawa@kf7.so-net.ne.jp (T.N.); t@ichou.com (O.T.); shungtang@hotmail.com (S.Y.); kouchiei@gmail.com (C.U.); kurokawak-int@h.u-tokyo.ac.jp (K.K.); miho.littlefield@hotmail.com (M.O.); tyism123@gmail.com (T.Y.); 2Department of Gastroenterology and Hepatology, International University of Health and Welfare, Narita Hospital, Narita 286-8520, Japan; mb99md@gmail.com (M.M.); ebinuma@me.com (H.E.); 3Department of Gastroenterology, Graduate School of Medicine, The University of Tokyo, Tokyo 113-8655, Japan; mtfujish@gmail.com; 4Division of Gastroenterology and Hepatology, Department of Internal Medicine, Tokai University School of Medicine, Isehara 259-1193, Japan

**Keywords:** TXI, sessile serrated lesion, hyperplastic polyp, colonoscopy

## Abstract

Background and aim: Olympus Corporation released the texture and color enhancement imaging (TXI) technology as a novel image-enhancing endoscopic technique. We investigated the effectiveness of TXI in the imaging of serrated colorectal polyps, including sessile serrated lesions (SSLs). Methods: Serrated colorectal polyps were observed using white light imaging (WLI), TXI, narrow-band imaging (NBI), and chromoendoscopy with and without magnification. Serrated polyps were histologically confirmed. TXI was compared with WLI, NBI, and chromoendoscopy for the visibility of the lesions without magnification and for that of the vessel and surface patterns with magnification. Three expert endoscopists evaluated the visibility scores, which were classified from 1 to 4. Results: Twenty-nine consecutive serrated polyps were evaluated. In the visibility score without magnification, TXI was significantly superior to WLI but inferior to chromoendoscopy in the imaging of serrated polyps and the sub-analysis of SSLs. In the visibility score for vessel patterns with magnification, TXI was significantly superior to WLI and chromoendoscopy in the imaging of serrated polyps and the sub-analysis of SSLs. In the visibility score for surface patterns with magnification, TXI was significantly superior to WLI but inferior to NBI in serrated polyps and in the sub-analysis of SSLs and hyperplastic polyps. Conclusions: TXI provided higher visibility than did WLI for serrated, colorectal polyps, including SSLs.

## 1. Introduction

Globally, colorectal cancer is the third most diagnosed malignancy [1]. The endoscopic resection of colorectal polyps could reduce colorectal cancer mortality by over 50%, providing evidence for the importance of endoscopic resection [2,3].

Recently, the serrated polyp–cancer sequence has received considerable attention, and it is responsible for up to 20% of all sporadic colorectal cancers. Serrated polyps are classified into three categories: hyperplastic polyps (HPs), sessile serrated lesions (SSLs), and traditional serrated adenomas [4]. Of these, SSL and traditional serrated adenoma are both precursors of cancer [5]. SSL predominantly occurs in the right side of the colon and is associated with B-RAF mutation and high microsatellite instability [6]. SSL is often difficult to detect because it typically has indistinct borders, and the color is similar to the background mucosa or is slightly faded [7]. SSL is often overlooked, though it accounts for a significant proportion of interval cancers [8]. Thus, it is desirable to improve the detection sensitivity for SSL.

Recently, Olympus Corporation released texture and color enhancement imaging (TXI) as an image-enhanced endoscopy technology in the new endoscopy system EVIS X1. Briefly, TXI enhances texture and color and adjusts brightness. TXI consists of six consecutive processes: (i) The input image is split into a base layer and detail layer. (ii) The brightness in the dark regions of the base layer is adjusted. (iii) Tone-mapping is applied to the corrected base layer. (iv) Texture enhancement is applied to the detail layer to enhance the subtle contrast. (v) The base layer after tone-mapping and the detail layer after texture enhancement are recombined. A TXI image produced in the fifth step is TXI mode 2 (texture and brightness enhancement). (vi) Color enhancement is applied to the output of TXI mode 1 to more clearly define the slight color contrast. The color enhancement algorithm of TXI was designed to expand the color difference between red and white hues in the image. TXI more greatly improves the visibility for colorectal adenoma than does white light imaging (WLI). In this study, we investigate the effectiveness of TXI for colorectal serrated polyps, especially SSLs.

## 2. Methods

### 2.1. Patients

We enrolled patients who underwent endoscopic resection for serrated colorectal lesions at Toyoshima Endoscopy Clinic from March to June 2021. When colorectal lesions were endoscopically diagnosed or were suspected to be SSLs, they were removed. When patients had multiple polyps, the polyps were treated individually. All of the resected specimens were examined histologically under hematoxylin and eosin staining. Indications for colonoscopy included the evaluation for symptoms, examination for a positive fecal occult blood test, polyp surveillance, and a medical check-up [9].

### 2.2. Ethics

This study was conducted in accordance with the ethical guidelines of the Declaration of Helsinki. This study was approved by the Certificated Review Board of Yoyogi Mental Clinic on 16 July 2021 (approval no. RKK227).

### 2.3. Endoscopy

We used the EVIS X1 video system center (CV-1500), a 4K resolution ultra-high-definition liquid crystal display monitor (OEV321UH), and a CF-HQ290Z colonoscope (Olympus Corp., Tokyo, Japan). For chromoendoscopy, we used 0.05% indigo carmine [10].

One expert endoscopist performed the colonoscopy and observation using the WLI, TXI, narrow band imaging (NBI), and chromoendoscopy modalities [11]. Lesions were first washed carefully with water to remove the mucus. Images were then obtained through WL, TXI, and NBI with distanced observation, followed by magnified observation. The lesions were subsequently stained for chromoendoscopy with indigo carmine, and images were obtained with and without magnification. TXI has mode 1 and mode 2. Mode 2 involves texture enhancement and brightness adjustment, and mode 1 adds color enhancement to mode 2. Mode 1 was used in this study.

### 2.4. Visibility Scoring

We investigated the visibility of the lesions, the vessel patterns, and surface patterns.

The visibility of the lesions was defined as the detectability of the lesions without magnification. The visibility of the vessel patterns was defined as the visibility of micro-vessels and varicose microvascular vessels using magnification. The visibility of the surface patterns was defined as the visibility of the mucosal structure, including the white zone, pit, and expanded crypt opening, using magnification. An expanded crypt opening is a feature of SSLs.

As in previous reports, the visibility score was defined as follows: 4, excellent (easily detectable); 3, good (detectable with careful observation); 2, fair (hardly detectable without careful examination); and 1, poor (not detectable without repeated careful examination) [12,13]. Representative images of each score are shown in Figure 1 and Figure 2. Three expert endoscopists evaluated the visibility scores.

### 2.5. Statistical Analysis

The continuous variables are expressed as mean ± standard deviation (SD). Continuous data between the four groups were compared using Dunn’s test with the Kruskal–Wallis test. Continuous data between the two groups were compared using the signed-rank test. Calculations were performed using Stat Mate IV software (version 4.01, ATOMS, Tokyo, Japan). Statistical significance was defined as a *p* < 0.05.

## 3. Results

### 3.1. Patients

Table 1 shows the clinicopathological characteristics of the 27 consecutive patients with 29 serrated polyps evaluated in this study. Histologically, there were 18 SSLs and 11 microvesicular mucin-rich type hyperplastic polyps.

### 3.2. Visibility Score for the Lesion without Magnification

The lesion visibility score of TXI was significantly higher than that of WLI but lower than that of chromoendoscopy in serrated polyps and the sub-analysis of SSLs (Table 2).

### 3.3. Visibility Score for Vessel Pattern with Magnification

The visibility score of TXI for the vessel pattern with magnification was significantly higher than that of WLI and chromoendoscopy in serrated polyps and the sub-analysis of SSLs (Table 3).

### 3.4. Visibility Score for Surface Pattern with Magnification

The visibility score of TXI for the surface pattern with magnification was significantly higher than that of WLI but lower than that of NBI in serrated polyps and the sub-analysis of SSLs and hyperplastic polyps (Table 3).

### 3.5. Visibility Scores for WLI and TXI by Each Expert Endoscopist

The visibility scores of TXI with and without magnification were significantly higher than those of WLI (Table 4). The visibility improvement in TXI was consistent among the three expert endoscopists.

## 4. Discussion

This study showed that TXI with and without magnification provided higher visibility than WLI did for serrated colorectal polyps, including SSL. However, non-magnified TXI was inferior to chromoendoscopy, and magnified TXI for surface patterns was inferior to magnified NBI. This is the first report on the efficacy of TXI for serrated colorectal polyps, including SSL.

Olympus Corp. first developed the NBI as an innovative image-enhanced endoscopy technology in 2007 [14]. Fujifilm Corp. developed blue light imaging (BLI) as a similar product. NBI and BLI are mainly used for magnified endoscopy and the detection of esophageal cancer [15,16]. Fujifilm Corp. also developed the linked color imaging (LCI) method, which is mainly used to detect lesions without magnification [17]. A randomized controlled trial (RCT) showed that LCI was significantly superior to standard WLI for polyp detection [18]. Currently, LCI-based observations are becoming standard instead of WLI. However, Olympus did not have a mode corresponding to that of LCI until recently. Olympus released TXI as a mode similar to that of LCI. Although LCI and TXI have similar concepts, there are several differences in their principles. LCI uses ambient light with wavelengths of 410 nm and 450 nm, the images are converted to resemble those of WLI, and color is enhanced such that red is changed to vivid red and white is changed to clear white. On the other hand, TXI uses white light, texture and color are enhanced, and brightness is adjusted. There is no study that directly compares TXI and LCI. The comparison between TXI and LCI is a future issue. The global share of Olympus Corp. is 70% for gastrointestinal endoscopy, so the spread of TXI may exceed that of LCI.

TXI provides higher visibility than does WLI for colorectal adenomas. In the present study, we found that TXI provided higher visibility than did WLI for serrated colorectal polyps. Taken together, this might imply that TXI can replace WLI in the detection of premalignant polyps in colonoscopy.

It is controversial whether LCI allows for better SSL detection. Fujimoto et al. showed that LCI was the most sensitive mode for SSL detection among WLI, BLI, and LCI in still-image examinations. Furthermore, their RCT of tandem colonoscopy with WLI and LCI suggested that LCI is superior to WLI in SSL detection [19]. An RCT by Dos Santos showed that LCI enables better adenoma detection, with a borderline significance for a higher detection of sessile serrated adenomas (*p* = 0.05) [20]. Conversely, an RCT by Paggi et al. showed that LCI allowed for better adenoma detection, but not for SSL detection [21]. Our study showed that TXI was significantly superior to WLI for SSL detection in still-image examinations. With regard to the SSL detection rate during colonoscopy, prospective RCTs are required in the future.

In this study, TXI was inferior to chromoendoscopy in SSL detection. Furthermore, the magnified TXI was inferior to the magnified NBI in terms of the visibility of surface patterns. On the other hand, Kitagawa et al. found that magnified LCI with indigo carmine was superior to magnified BLI [22]. Sakamoto et al. also reported that magnified LCI with crystal violet staining provided more diagnostic information than magnified blue light imaging (BLI) and WLI [23]. TXI with chromoendoscopy might be also promising, and needs to be further investigated in future studies.

Texture plays an important role in the identification of regions of interest in an image; hence, texture enhancement is a meaningful component in digital image processing [24,25]. There are several reports on the quantitative analysis of TXI. Sato et al. performed a quantitative analysis using endoscopic images of the gastrointestinal tract from an in vivo porcine study [26]. Their quantitative analysis included edge-based contrast measurements, the standard deviation of the averaged illumination, and the color difference. In the analysis of edge-based contrast measure, TXI had higher value than WLI, showing that TXI can enhance image contrast, arising from texture enhancement. This study also revealed that TXI can reduced the standard deviation of the illumination nonuniformity compared with white light imaging (WLI). This improvement was achieved by selectively enhancing the brightness in dark areas. In the analysis of color difference, TXI had a higher color difference than WLI due to color enhancement. Ishikawa et al. also analyzed the color difference between neoplastic and peripheral areas of twelve gastric neoplasms in WLI and TXI [27]. The color difference was significantly higher in TXI than in WLI.

The present study had some limitations. The sample size was small, although statistically significant differences were observed. Larger prospective studies are required in the future. Even though this was a single-center, retrospective study, because our institution specializes in endoscopy, the endoscopic environment was well managed.

## 5. Conclusions

The visibility provided by non-magnified TXI was higher than that provided by WLI and lower than that provided by chromoendoscopy for serrated polyps, including SSLs. The visibility provided by magnified TXI was higher than that provided by WLI but lower than that provided by NBI. TXI could be a suitable modality for the detection of premalignant colorectal polyps.

## Figures and Tables

**Figure 1 jcm-11-00119-f001:**
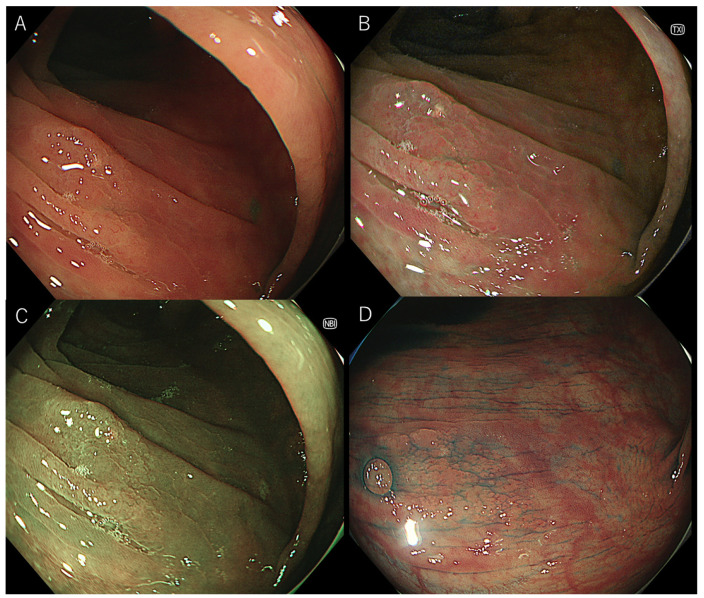
Representative images of a sessile serrated lesion without magnification. (**A**) white-light imaging, (**B**) texture and color enhancement imaging, (**C**) narrow-band imaging, (**D**) chromoendoscopy.

**Figure 2 jcm-11-00119-f002:**
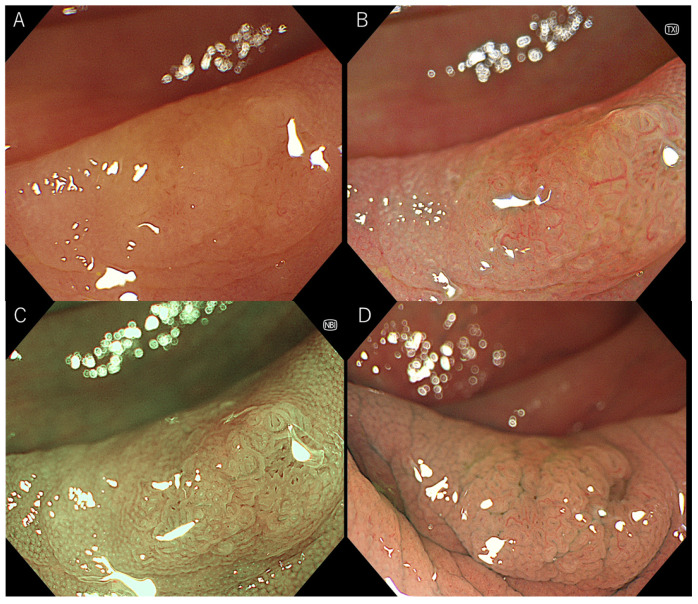
Representative images of sessile serrated lesion with magnification. (**A**) white-light imaging, (**B**) texture and color enhancement imaging, (**C**) narrow-band imaging, (**D**) chromoendoscopy.

**Table 1 jcm-11-00119-t001:** Clinicopathological characteristics of patients and adenomas.

Serrated Polyps, *n.*	29
Histological subtype; *n.*	
Sessile serrated lesion	18
Microvesicular mucin-rich type hyperplastic polyp	11
Goblet cell-rich type hyperplastic polyp	0
Location; *n.*, cecum, ascending, transverse, descending, sigmoid, rectum	4, 12, 10, 0, 3, 0
Size, mean (standard deviation, range); mm	9.0 (4.29, 3–18)
Morphology; *n.*, Ip, Is, IIa, IIb	0, 0, 29, 0

**Table 2 jcm-11-00119-t002:** Visibility scores without magnification for WLI, TXI, NBI, and chromoendoscopy.

	WLI	TXI	NBI	Chromoendoscopy
All serrated polyps				
Mean visibility score (SD)	2.27 (0.75)	2.93 (0.76) ***	2.74 (0.79) **	3.45 (0.68) ***, †††, ‡‡‡
SSLs				
Mean visibility score (SD)	2.25 (0.76)	2.90 (0.77) ***	2.65 (0.79)	3.45 (0.64) ***, ††, ‡‡‡
Hyperplastic polyps				
Mean visibility score (SD)	2.30 (0.72)	2.97 (0.67) **	2.88 (0.77) *	3.45 (0.74) ***, ‡

The visibility score was defined as follows: 4, excellent (easily detectable); 3, good (detectable with careful observation; 2, fair (hardly detectable without careful examination); and 1 poor (not detectable without repeated careful examination). WLI, white light imaging; TXI, texture and color enhancement imaging; NBI, narrow-band imaging; SD, standard deviation. ***: *p* value < 0.001 compared with WLI, **: *p* value < 0.01 compared with WLI, *: *p* value < 0.05 compared with WLI, †††: *p* value < 0.001 compared with TXI, ††: *p* value < 0.01 compared with TXI, ‡‡‡: *p* value < 0.001 compared with NBI, ‡: *p* value < 0.05 compared with NBI.

**Table 3 jcm-11-00119-t003:** Visibility scores of vessel pattern and surface pattern with magnification for WLI, TXI, NBI, and chromoendoscopy.

	WLI	TXI	NBI	Chromoendoscopy
All serrated polyps				
Vessel pattern (SD)	2.30 (0.74)	2.91 (0.80) ***, †††	3.23 (0.84) ***, †††	2.21 (0.83)
Surface pattern (SD)	1.86 (0.64)	2.75 (0.68) ***	3.46 (0.70) ***, ‡‡‡, †††	2.79 (0.75) ***
SSLs				
Vessel pattern (SD)	2.24 (0.74)	2.89 (0.81) ***, ††	3.19 (0.86) ***, †††	2.30 (0.87)
Surface pattern (SD)	1.80 (0.62)	2.70 (0.66) ***	3.39 (0.78) ***, ‡‡‡, ††	2.83 (0.76) ***
Hyperplastic polyps				
Vessel pattern (SD)	2.41 (0.73)	2.96 (0.79) †††	3.33 (0.77) ***, †††	2.04 (0.69)
Surface pattern (SD)	2.00 (0.67)	2.85 (0.70) **	3.59 (0.49) ***, ‡‡, †††	2.70 (0.71) *

WLI, white light imaging; TXI, texture and color enhancement imaging; NBI, narrow-band imaging; SD, standard deviation. ***: *p* value < 0.001 compared with WLI, **: *p* value < 0.01 compared with WLI, *: *p* value < 0.05 compared with WLI, †††: *p* value < 0.001 compared with chromoendoscopy, ††: *p* value < 0.01 compared with chromoendoscopy, ‡‡‡: *p* value < 0.001 compared with TXI, ‡‡: *p* value < 0.01 compared with TXI.

**Table 4 jcm-11-00119-t004:** Visibility scores for WLI and TXI assigned by each expert endoscopist.

	WLI	TXI
Mean visibility scores without magnification (SD)		
Expert endoscopist 1	2.29 (0.71)	2.75 (0.75) ***
Expert endoscopist 2	2.46 (0.83)	3.00 (0.77) ***
Expert endoscopist 3	2.07 (0.66)	3.04 (0.69) ***
Visibility scores of vessel pattern with magnification		
Expert endoscopist 1	1.85 (0.60)	2.37 (0.74) ***
Expert endoscopist 2	2.52 (0.85)	2.89 (0.70) **
Expert endoscopist 3	2.52 (0.58)	3.48 (0.58) ***
Visibility scores of surface pattern with magnification		
Expert endoscopist 1	1.62 (0.88)	2.48 (0.80) ***
Expert endoscopist 2	1.89 (0.51)	2.74 (0.59) ***
Expert endoscopist 3	2.07 (0.38)	3.04 (0.52) ***

WLI, white light imaging; TXI, texture and color enhancement imaging; SD, standard deviation. ***: *p* value < 0.001 compared with WLI, **: *p* value < 0.01 compared with WLI.

## Data Availability

No additional data are available.

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
