# Peer review of "TXI (Texture and Color Enhancement Imaging) for Serrated Colorectal Lesions"

_jcm, 2021, doi:10.3390/jcm11010119_

Round 1
Reviewer 1 Report
This manuscript “Texture and color enhancement imaging (TXI) for serrated colorectal lesions” explored the effectiveness of TXI in imaging serrated colorectal polyps by observing the lesions, the vessel and surface patterns under TXI, WLI, NBI and chromoendoscopy. It may provide support for TXI pre-cancerous colorectal polyp detection. However, this manuscript cannot be published in its current form. I recommend that authors edit the manuscript while considering the following points:
- Isit possible to analyze the differences, that come form the three expert endoscopists. In order to show that the the TXI have a better stability with different physicians, and how easy or difficult the procedure is?.
- The manuscript uses visibility scores to evaluate TXI, WLI, NBI and chromoendoscopy for the diagnosis of serrated colorectal polyps. In the sub-analysis of SSL and hyperplastic polyps, TXI scored lower than NBI for the surface pattern with magnification. I wandered is there any other methods to enhance the effectiveness of TXI to observe lesions and vessel, surface pattern? Maybe a brief introduction about these methods could be added to the manuscript.
- The discussion section, TXI is a similar model to LCI, but with some differences in principle. The advantages of the TXI could be added.
Author Response
Thank you for your important comments, which were extremely helpful for improving the quality of our manuscript.
According to your comment, we analyzed the differences, that come from the three expert endoscopists. Table 4 was added.
Table 4 Visibility scores for WLI and TXI by each expert endoscopist
|
|
WLI |
TXI |
|
Mean visibility scores without magnification (SD) |
|
|
|
Expert endoscopist 1 |
2.29 (0.71) |
2.75 (0.75) *** |
|
Expert endoscopist 2 |
2.46 (0.83) |
3.00 (0.77) *** |
|
Expert endoscopist 3 |
2.07 (0.66) |
3.04 (0.69) *** |
|
Visibility scores of vessel pattern with magnification |
|
|
|
Expert endoscopist 1 |
1.85 (0.60) |
2.37 (0.74) *** |
|
Expert endoscopist 2 |
2.52 (0.85) |
2.89 (0.70) ** |
|
Expert endoscopist 3 |
2.52 (0.58) |
3.48 (0.58) *** |
|
Visibility scores of surface pattern with magnification |
|
|
|
Expert endoscopist 1 |
1.62 (0.88) |
2.48 (0.80) *** |
|
Expert endoscopist 2 |
1.89 (0.51) |
2.74 (0.59) *** |
|
Expert endoscopist 3 |
2.07 (0.38) |
3.04 (0.52) *** |
WLI, white light imaging; TXI, texture and color enhancement imaging; SD, standard deviation. ***: P value <0.001 compared with WLI, **: P value <0.01 compared with WLI
The visibility scores of TXI with and without magnification were significantly higher than those of WLI. The visibility improvement in TXI was consistent with 3 expert endoscopists. This point was added into the revised manuscript (page 5 line 23 – page 6 line 1; page 6 line 19 - page 7 line 3; Table 4).
Sakamoto et al. reported that magnified LCI with crystal violet staining provided more diagnostic information than magnified blue light imaging (BLI) and WLI (Int J Colorectal Dis 2019; 34(7): 1341-1344). TXI with chromoendoscopy might be also promising, and needs to be further investigated in future studies. This point was added into the revised manuscript (page 8 line 19-21).
According to your comment, the introduction about these methods was added into the introduction section as follows (page 3 line 18 – page 4 line 2).
“TXI consists of six consecutive processes: (i) The input image is split into base layer and detail layer. (ii) The brightness in the dark regions of the base layer is adjusted. (iii) Tone-mapping is applied to the corrected base layer. (iv) Texture enhancement is applied to the detail layer to enhance the subtle contrast. (v) The base layer after tone-mapping and the detail layer after texture enhancement are recombined. A TXI image produced in the fifth step is TXI mode 2 (texture and brightness enhancement). (vi) Color enhancement is applied to the output of TXI mode 1 to more clearly define the slight color contrast. The color enhancement algorithm of TXI was designed to expand the color difference between red and white hues in the image”
There is no study that directly compares TXI and LCI. The comparison between TXI and LCI is future issue. The advantage of the TXI might depend on the Olympus brand. The global share of Olympus is 70% for gastrointestinal endoscopy. Therefore, the spread of TXI may exceed that of LCI. This point was added into the revised manuscript (page 7 line 23 - page 8 line 1).

Reviewer 2 Report
"Texture and color enhancement imaging (TXI) was developed as a novel image-enhancing endo- 19 scopic technique by Olympus Corp"
There is information lacking about technical information desribing type of image processing used to influence te image recived by operator.
There is no information on what basis authors state that image was improved - enriched colors, scharp edges, better resolution (?) aforementioned criteria supposed to be analysed separetly and estimateted statristically.
"One expert endoscopist performed the colonoscopy and observation using the WLI, 77 TXI, narrow band imaging (NBI), and chromoendoscopy modalities"
Analysis made by one endoscopist seems very shallow what might be sufficient for preliminary report for the conference but not for scientific article.
Twenty-nine consecutive serrated polyps were evaluated
Group of 29 patients seems to be not sufficient as I - each type of polyp supposed to be analysed in the group of minimum 30 images therefore group of patients supposed to be significantly larger.
"We investigated the visibility of the lesions, the vessel patterns, and surface patterns"
What type of surface patterns were analysed ? I think that according to thi article title sugesting texture enchancemet is very misleading. Textures are group of image features and can not be enchanced - can be processed and analysed
moreover technique of the textural analysis was not introduced by Olympus (!) - texture analysis I mean) but was introducted as Image [processing mathematical method in early 1972 this has to be mentioned here.
Results section is very short some results are in the discussion section presented . There is lacking discussion with state of the art in the field.
Author Response
Thank you for your insightful comments, which were extremely helpful for improving the quality of our manuscript.
According to your comment, the information about these methods was added into the introduction section as follows (page 3 line 18 – page 4 line 2).
“TXI (texture and color enhancement imaging) consists of six consecutive processes: (i) The input image is split into base layer and detail layer. (ii) The brightness in the dark regions of the base layer is adjusted. (iii) Tone-mapping is applied to the corrected base layer. (iv) Texture enhancement is applied to the detail layer to enhance the subtle contrast. (v) The base layer after tone-mapping and the detail layer after texture enhancement are recombined. A TXI image produced in the fifth step is TXI mode 2 (texture and brightness enhancement). (vi) Color enhancement is applied to the output of TXI mode 1 to more clearly define the slight color contrast. The color enhancement algorithm of TXI was designed to expand the color difference between red and white hues in the image”
As you pointed out, there was no information on what basis authors state that image was improved. Sato et al. performed a quantitative analysis using endoscopic images of the gastrointestinal tract from an in vivo porcine study (J Healthc Eng 2021;2021:5518948). The quantitative analysis included standard deviation of averaged illumination, color difference, and edge-based contrast measure (EBCM). This study revealed that TXI can reduced the standard deviation of the illumination nonuniformity compared with white light imaging (WLI). This improvement was achieved by selectively enhancing the brightness in dark areas. In the analysis of color difference, TXI had a higher color difference than WLI due to color enhancement. In the analysis of EBCM, TXI had higher value than WLI showing that TXI can enhance image contrast arising from texture enhancement. Ishikawa et al. also analyzed the color difference between neoplastic and peripheral areas of 12 gastric adenocarcinomas and adenomas in WLI and TXI (Sci Rep2021;11: 6910). In this study, researchers individually entered points for inner and peripheral aspects of each image. The color differences were calculated using the CIE L*a*b* color space system (J Opt Soc Am1976;66: 497–500). The color difference was significantly higher in TXI than in WLI. These points were added into the revised manuscript (page 8 line 23 - page 9 line 11).
  One expert endoscopist performed the colonoscopy and observation using the WLI, TXI, narrow band imaging (NBI), and chromoendoscopy modalities. Three expert endoscopists evaluated the visibility score. we analyzed the differences, that come from the three expert endoscopists. Table 4 was added.
Table 4 Visibility scores for WLI and TXI by each expert endoscopist
|
|
WLI |
TXI |
|
Mean visibility scores without magnification (SD) |
|
|
|
Expert endoscopist 1 |
2.29 (0.71) |
2.75 (0.75) *** |
|
Expert endoscopist 2 |
2.46 (0.83) |
3.00 (0.77) *** |
|
Expert endoscopist 3 |
2.07 (0.66) |
3.04 (0.69) *** |
|
Visibility scores of vessel pattern with magnification |
|
|
|
Expert endoscopist 1 |
1.85 (0.60) |
2.37 (0.74) *** |
|
Expert endoscopist 2 |
2.52 (0.85) |
2.89 (0.70) ** |
|
Expert endoscopist 3 |
2.52 (0.58) |
3.48 (0.58) *** |
|
Visibility scores of surface pattern with magnification |
|
|
|
Expert endoscopist 1 |
1.62 (0.88) |
2.48 (0.80) *** |
|
Expert endoscopist 2 |
1.89 (0.51) |
2.74 (0.59) *** |
|
Expert endoscopist 3 |
2.07 (0.38) |
3.04 (0.52) *** |
WLI, white light imaging; TXI, texture and color enhancement imaging; SD, standard deviation. ***: P value <0.001 compared with WLI, **: P value <0.01 compared with WLI
The visibility scores of TXI with and without magnification were significantly higher than those of WLI. The visibility improvement in TXI was consistent with 3 expert endoscopists. This point was added into the revised manuscript (page 5 line 23 – page 6 line 1; page 6 line 19 - page 7 line 3; Table 4).
As you pointed out, the sample size was small in this study. Larger prospective studies are required in the future. This point was added into the revised manuscript (page 9 line 12-14).
The visibility of the surface patterns was defined as the visibility of mucosal structure, including the white zone, pit, and expanded crypt opening using magnification. Especially, expanded crypt opening is the feature of sessile serrated lesions (SSLs). This point was added into the revised manuscript (page 5 line 14).
As you pointed out, the article title could be misleading. Therefore, the title was modified to “TXI (texture and color enhancement imaging) for serrated colorectal lesions”.
As you pointed out, texture analysis has a long history. Technique of the textural analysis was not introduced by Olympus. Texture plays an important role in the identification of regions of interest in an image, hence texture enhancement is a meaningful component in digital image processing. The manuscript was modified, and reports about textural analysis were cited into the revised manuscript (page 8 line 23-24).
Owing to you and reviewer 1, the discussion was enriched. Thank you very much.
